# Impact of Partnered Pharmacist Medication Charting (PPMC) on Medication Discrepancies and Errors: A Pragmatic Evaluation of an Emergency Department-Based Process Redesign

**DOI:** 10.3390/ijerph20021452

**Published:** 2023-01-13

**Authors:** Tesfay Mehari Atey, Gregory M. Peterson, Mohammed S. Salahudeen, Luke R. Bereznicki, Tom Simpson, Camille M. Boland, Ed Anderson, John R. Burgess, Emma J. Huckerby, Viet Tran, Barbara C. Wimmer

**Affiliations:** 1School of Pharmacy and Pharmacology, College of Health and Medicine, University of Tasmania, Hobart 7005, Australia; 2Pharmacy Department, Royal Hobart Hospital, Tasmanian Health Service, Hobart 7000, Australia; 3Department of Endocrinology, Royal Hobart Hospital, Tasmanian Health Service, Hobart 7000, Australia; 4Tasmanian School of Medicine, College of Health and Medicine, University of Tasmania, Hobart 7000, Australia; 5Emergency Department, Royal Hobart Hospital, Tasmanian Health Service, Hobart 7000, Australia; 6Menzies Institute for Medical Research, College of Health and Medicine, University of Tasmania, Hobart 7000, Australia

**Keywords:** pharmacist, co-charting, medication charting, PPMC, emergency department, medication discrepancy, medication error

## Abstract

Medication errors are more prevalent in settings with acutely ill patients and heavy workloads, such as in an emergency department (ED). A pragmatic, controlled study compared partnered pharmacist medication charting (PPMC) (pharmacist-documented best-possible medication history [BPMH] followed by clinical discussion between a pharmacist and medical officer to co-develop a treatment plan and chart medications) with early BPMH (pharmacist-documented BPMH followed by medical officer-led traditional medication charting) and usual care (traditional medication charting approach without a pharmacist-collected BPMH in ED). Medication discrepancies were undocumented differences between medication charts and medication reconciliation. An expert panel assessed the discrepancies’ clinical significance, with ‘unintentional’ discrepancies deemed ‘errors’. Fewer patients in the PPMC group had at least one error (3.5%; 95% confidence interval [CI]: 1.1% to 5.8%) than in the early BPMH (49.4%; 95% CI: 42.5% to 56.3%) and usual care group (61.4%; 95% CI: 56.3% to 66.7%). The number of patients who need to be treated with PPMC to prevent at least one high/extreme error was 4.6 (95% CI: 3.4 to 6.9) and 4.0 (95% CI: 3.1 to 5.3) compared to the early BPMH and usual care group, respectively. PPMC within ED, incorporating interdisciplinary discussion, reduced clinically significant errors compared to early BPMH or usual care.

## 1. Introduction

A hospital’s emergency department (ED) serves as a vital entry point into the health care system for acutely ill patients, especially the most vulnerable individuals. Due to the high patient volume, overcrowding, emergent nature of care, complex undifferentiated medical issues, and often inadequate information structure, the ED setting is more prone to unintended medication regimen changes than other departments of the hospital [1,2,3,4]. The prevalence of medication errors, defined as “preventable events leading to inappropriate medication use or patient harm” [5], may range from 13.5% to 59.4% of patients in the ED [6,7,8,9].

Around half of the medication errors (47.8%) identified in the ED may be potentially serious [10]. If undetected early, 52.4% of medication errors may necessitate later intervention or intensive monitoring [11]. It is worth noting that the Australian Health Ministers recently agreed to recognise medicine safety and quality use of medicines as one of the national health priority areas [12,13]. The World Health Organisation also issued *The Global Patient Safety Challenge: Medication without Harm* in 2017, aiming for a 50% reduction in severe, preventable medication-related harms over five years [14].

In a recent trial in Victoria (Australia), partnered pharmacist medication charting (PPMC) showed promising results in patients with complex medication regimens or polypharmacy in a general medicine unit and an emergency short-stay unit [15]. A similar PPMC model of care was trialed in Tasmania’s Royal Hobart Hospital (RHH) ED, targeting adult patients (≥18 years) presenting to ED with subsequent admissions to a general medicine unit, emergency medicine unit, or mental health unit. This study aimed to evaluate the impact of PPMC on the prevalence of medication discrepancies and errors, and their clinical significance. In addition, this study investigated whether obtaining and documenting an early best-possible medication history (BPMH) in ED, without PPMC, could reduce medication discrepancies and errors compared to usual care.

## 2. Materials and Methods

### 2.1. Partnered Pharmacist Medication Charting

The Tasmanian Government Department of Health approved a 12 month PPMC project in the RHH ED in May 2020. The hospital, located in southern Tasmania (Australia), provides services to approximately a quarter-million people each year, with over 63,000 annual ED visits [16]. The project was funded by the Tasmanian Government, with the evaluation being outsourced to an independent research group (the University of Tasmania [UTAS]). A multidisciplinary RHH PPMC working group, comprised of 16 representatives from ED, acute medical specialty units, nursing, critical care, medical services, pharmacy, and academia, was established to oversee the project. The working group, guided by a previously published approach [17], developed a PPMC model of care specific to the needs of the Tasmanian Health Service (THS). The final PPMC model was approved by the THS Medical Leadership Advisory Council and then by the THS Policy Committee, prior to its implementation.

Over 20 pharmacists were trained and credentialled to provide PPMC seven days a week, in shifts from 8 am to 8 pm, in the PPMC’s early stages, and 8 am to 9 pm after two months. The following were PPMC-credentialling pathway requirements: (a) completion of all prior reading and education modules, (b) clinical assessment via the Society of Hospital Pharmacists of Australia (SHPA)’s Clinical Competency Achievement Tool (ClinCAT) [18], (c) observation of one PPMC clinical episode, (d) completion of five PPMC clinical episodes under the supervision of a PPMC-credentialled pharmacist or a medical officer, (e) completion of a PPMC-Training Objective Structured Clinical Examination with a supervising pharmacist and a medical consultant, and (f) formal recognition by the THS State-wide Credentialling Committee. ClinCAT^®^ is a competency achievement tool that supports Australian pharmacists’ training and development. Accredited by the Australian Pharmacy Council, ClinCAT^®^ includes activities defined by SHPA’s standards of practice for clinical pharmacy services [19].

### 2.2. Study Design, Population, and Period

A controlled concurrent pragmatic evaluation was conducted among people aged 18 years or older, presenting to the ED between 1 June 2020 and 17 May 2021. A comparison was made between three different study arms: the PPMC arm (i.e., a process ‘redesign’), the early BPMH arm (i.e., a process ‘tweak’), and the usual care arm (i.e., a traditional standard of care) (Appendix A).

### 2.3. Assignment of Patients into Study Arms, and Inclusion and Exclusion Criteria

Based on the PPMC model, patients were pragmatically assigned to one of the study arms by ED staff as part of their routine patient care (Appendix A). ED staff stratified patients into ‘low-risk’ or ‘medium-high risk’ (i.e., 65 years or above, taking any prescribed preadmission medication, or having a mental health admission, questionable adherence, or any medication concerns) using a patient risk assessment framework. Only medium-high risk patients, who were intended to be admitted through ED to one of the three acute medical units (AMUs), i.e., either a general medicine, emergency medicine, or mental health unit, were eligible for inclusion in the PPMC arm. AMUs consisted of a wide range of medical specialties that provide care for patients admitted primarily for medical reasons. An emergency medicine unit is an extension to ED that provides care for patients expecting to stay less than 24 h in the hospital, without the need for specialist inpatient management. Additional AMU admissions via ED, such as cardiology, respiratory, renal, rheumatology, endocrinology, or stroke unit, were included in the early BPMH arm or the usual care arm. If ED medical officers had already charted patients’ medications, these patients were ineligible for inclusion in either the PPMC arm or the early BPMH arm and were instead included in the usual care arm. Additionally, on occasion, the PPMC pharmacists’ workloads led to an ‘overflow’ of patients into the early BPMH arm or the usual care arm (which occurred in about 6% of patients intended for the PPMC arm).

Adults, aged 18 years or above, presenting to ED with subsequent planned admission to one of the AMUs, taking at least one regular preadmission medication and receiving their first medication reconciliation (MedRec) on the ward within 48 h of ED transfer, were included in the study. MedRec is defined as “the formal process of obtaining and verifying a complete and accurate list of each patient’s current medicines and matching the medicines the patient should be prescribed to those they are actually prescribed in the hospital” [20]. Patients were excluded if they were children (younger than 18 years old), were not subsequently admitted to the hospital, were admitted to units other than an eligible AMU, had overnight ED presentations (9 pm to 8 am), lacked BPMH data, did not receive MedRec within 48 h after ED transfer (i.e., no MedRec conducted or MedRec conducted beyond the 48 h), or had incomplete data. Only the first eligible hospital admission, via ED, for each patient during the study period was included.

### 2.4. Study Arms

In the PPMC arm, a BPMH was first documented by an ED pharmacist at the earliest possible time point in the ED. The BPMH is a systematic process of obtaining a medication history that includes a thorough list of all medicines the patient is taking or has recently used, determined through a structured patient interview and secondary sources, such as caregivers, national prescription and dispensing repository, electronic health records, hospital digital medical records, community pharmacies, and/or residential care facilities [20]. After conducting a clinical review, a PPMC-credentialled pharmacist subsequently had a conversation with a medical officer (at least a post-graduate year 2 resident) to jointly develop a shared medication treatment plan (SMTP). The SMTP included agreements on therapy to be documented in the initial medication chart. In the approved THS-specific PPMC model, the medications could only be charted by the PPMC-credentialled pharmacist, using a purple ink ballpoint pen, after the conversation and completion of the co-signed SMTP. The medical officer then endorsed the medications charted before the nursing staff administered them. A ward pharmacist subsequently conducted a MedRec on the inpatient ward, in which usual clinical pharmacy services were provided.

The early BPMH arm included documentation of a BPMH by an ED pharmacist as early as feasible in the ED. This was followed by a traditional medication charting approach, in which a medical officer wrote medication charts in the ED using a black/blue ink ballpoint pen. In this arm, there was no clinical discussion between the PPMC pharmacist and the medical officer in ED; the BPMH was available to the medical officer prior to charting. Afterwards, a ward pharmacist performed a MedRec on the inpatient ward. The usual care arm included the traditional medication charting without a pharmacist-collected BPMH in ED, followed by a ward pharmacist-led MedRec on the inpatient ward.

### 2.5. Data Collection

From October 2020 to December 2021, a non-blinded, independent researcher (TMA) retrospectively collected data by linking multiple datasets (i.e., ED presentation, MedRec, BPMH, admission, and PPMC data) and accessing patients’ digital medical records. Data were also retrospectively collected from the Healthcare Software Clinical Suite, a system that enables clinical pharmacists to record a patient’s BPMH and MedRec in the form of a medication management action plan (MedMAP).

Each patient’s regular, as-needed (*Pro Re Nata* [PRN]), and complementary medicines were collected using data documented in the BPMH, initial medication charts, MedRec, and ED and hospital discharge summaries/prescriptions. The Australasian Triage Scale (ATS) was used to categorise the acuity of ED presentations, with a score of 1 indicating the most critical and 5 indicating the least critical [21]. An age-adjusted Charlson comorbidity index (CCI), which factors in 17 medical disorders and the patient’s age, was used to assess the extent of comorbidities [22]. 

### 2.6. Outcome Measures

The primary outcome was the percentage of patients having at least one medication error. Secondary outcomes included types, prevalence, and intentionality of medication discrepancies, likely clinical significance of medication errors and the number of medication errors per prescribed medication. Medication errors involving pre-defined high-risk medicines (HRMs) and time-critical medicines (TCMs) were also reported. Drugs in the APINCH classification (Antimicrobials, Potassium and other electrolytes [injections], Insulin, Narcotics [opioids] and other sedatives, Chemotherapeutic agents, and Heparin and other anticoagulants) [23] and those with a narrow therapeutic index were included as HRMs (Appendix A). TCMs are defined as “medicines where early or delayed administration by more than 30 min from the prescribed time for administration may cause harm to the patient or compromise the therapeutic effect” [24]. A local THS list was used for the TCMs (Appendix A).

A review of the available literature indicated that medication discrepancies and errors are characterised in several ways [25]. In the absence of a gold-standard definition, and for the purpose of reproducibility, an in-house classification system was developed considering the project’s core purpose and literature [26,27,28,29]. The UTAS research team and the RHH PPMC working group reached a consensus on the pre-determined definitions, categories, and examples of medication discrepancies/errors (Appendix A).

For the purpose of this study, an undocumented medication discrepancy (henceforth ‘discrepancy’) was defined as any difference, without a documented explanation, between the actual inpatient medication chart and the first MedRec (intended to reflect what the patient should have been taking), undertaken within 48 h of transfer from ED by the ward clinical pharmacists. A medication error (henceforth ‘error’) was defined as any discrepancy that was deemed ‘unintentional’ by an expert panel in the context of a patient’s acute presentation. The definitions were limited to discrepancies/errors involving all regular, including medically indicated vitamins, or complementary medicines (e.g., cholecalciferol for osteoporosis) and prescription as-needed (PRN) medications. 

During their everyday tasks, independent RHH ward-based clinical pharmacists prospectively identified medication charting and clinical issues and documented them in the MedMAP. The charting issues were retrospectively extracted from the MedMAP, based on the agreed-upon in-house definition, by the independent researcher (TMA). Patient cases were independently prepared (by TMA) and then blindly reviewed by each of the rest of the research team (BCW, GMP, MSS, and LRB) for completeness, clarity, and conciseness (Appendix A). The cases included all identified discrepancies in the PPMC group and approximately 10% of the discrepancies from each of the early BPMH group and the usual care group, based on a previous study [30]. The 10% discrepancies were randomly selected from each discrepancy type using a proportionate random sampling technique. The cases were evaluated in three rounds by five blinded, independent multidisciplinary experts—consisting of senior clinical pharmacists and medical consultants from each of emergency medicine and general medicine—using an SHPA standardised matrix tool [31]. The panel members were neither part of the PPMC pharmacy team nor PPMC working group.

Three multidisciplinary expert panel members independently assessed the cases in the first round. Each panel member first inferred the *intentionality* of each given discrepancy as being an “undocumented but likely-intentional discrepancy” or “unintentional medication error” (Figure 1). The likely clinical *risk* of errors was then determined from the degree of *severity* and *likelihood* of the *consequence* occurring, using the SHPA tool [31]. A fourth and fifth blinded independent senior clinician assessed significant differences from the previous round(s) in the second round and the third round, respectively. Significant differences were characterised as either (a) different assessments of *intentionality* (e.g., intentional-error-error) followed by different clinical *risk* ratings (e.g., moderate-high) or (b) similar assessments of *intentionality* (e.g., error-error-error) followed by one non-adjacent *risk* rating (e.g., low-low-high). At least two responses were similar or adjacent for all *risk* ratings and there were no instances of “low-moderate-extreme” ratings.

Initially, each panel member inferred the intentionality of each given discrepancy and classified it as either an “undocumented but likely intentional discrepancy” or an “unintentional medication error” (Step 1). The panel member then nominated the most severe (i.e., worst) possible clinical consequence, from their clinical experience, which could potentially occur as a result of the error irrespective of time (Step 2). Following this, the panel member determined the degree of severity, i.e., the magnitude of potential patient harm, which might have resulted from the consequence on a scale of 1 to 5 (insignificant [1], minor [2], moderate [3], major [4], or catastrophic [5]) had the error not been detected and corrected (Step 3). Finally, the panel member determined the likelihood of the consequence occurring on a probability scale of almost certain, likely, possible, unlikely, or rare, irrespective of time (Step 4).

The UTAS research team collated the panel assessment results in each round and then completed the risk matrix using the SHPA tool [31]. The research team did not make any judgment on the results and instead, where feasible, either compiled a majority vote or averaged panel responses when there was a disagreement (e.g., “moderate” for “moderate-moderate-high” or “low-moderate-high”).

### 2.7. Data Analysis

Data were analysed using R^®^ (V.4.1.12) (R Foundation for Statistical Computing, Vienna, Austria) [32]. Frequencies, percentages, medians (interquartile range [IQR]), and cross-tabulations were used for the descriptive analysis. Testing for normality of continuous variables was performed using the Shapiro-Wilk test and graphical methods. Categorical patient characteristics were compared with Pearson’s chi-square or Fisher’s exact tests, as appropriate. Non-parametric data were compared using the Kruskal-Wallis test, with Dunn’s for the post hoc test. All tests were two-tailed, and significance was set at *p* < 0.05. *p*-values were adjusted for multiple comparisons using the Benjamini-Hochberg method. 

Medication discrepancies/errors were reported as the percentage of patients having at least one discrepancy/error and the number of discrepancies/errors per 100 medication orders for the initially charted medicines (determined as the total number of discrepancies/errors in each group divided by the total number of medicines in each group and then multiplied by 100). Discrepancies were adjusted for each patient’s medications by dividing the number of discrepancies by the number of medicines for each patient. For each study group, the median of this ratio was determined and reported as ‘the median number of discrepancies per medication’. Errors bearing high risk or extreme risk were grouped as ‘high/extreme risk errors’. 

The number of patients who need to be treated (NNT) with PPMC to prevent at least one additional error was calculated as the reciprocal of absolute risk reduction (ARR). The ARR was estimated as the absolute difference in the percentage of patients having at least one error between the PPMC group and the comparison groups. The prevalence and clinical severity of the errors with their 95% uncertainties were extrapolated from the panel assessment findings. Ninety-five percent confidence intervals (CI) were estimated using a point proportion population formula ($p\;\pm \;z\;\sqrt{\frac{p\;(1-p)}{n}}$; where *p* was the sample proportion, z was the critical value for the 95% confidence level (1.96), and *n* was the population size). Fleiss’ kappa statistics were used to determine the degree of inter-panel agreements [33]. 

### 2.8. Ethical Considerations

Prior to commencement, the study received ethics approval (H0018682) from the University of Tasmania Human Research Ethics Committee and site authorisation approval from the THS Research Governance Office.

### 2.9. Sample Size Calculation

The required sample size was estimated based on medication discrepancies and 95% power, 5% two-sided significance level, and two degrees of freedom using a multigroup goodness-of-fit test with two-by-three contingency tables in G*Power (V3.1.9.4, Westphalia, Germany). Using literature evidence, a sample size of 229 per study group was determined to detect a reduction of 15% in the percentage of patients having at least one medication discrepancy when pharmacists were deployed in ED [34]. Using an online random number generator (http://izmm.com/random.pl, accessed on 10 November 2020), samples of patients from each group were selected randomly.

## 3. Results

### 3.1. Patient Characteristics

Between 1 June 2020 and 17 May 2021, a total of 62,662 patients presented to the RHH ED (Figure 2). Of these, 58,549 presentations were excluded using predetermined criteria: children under the age of 18 (*n* = 8836, 15.1%), those not subsequently admitted to the hospital (*n* = 34,636, 59.2%), those admitted to units other than an eligible AMU (*n* = 5923, 10.1%), overnight ED presentations (*n* = 3273, 5.6%), not being the first eligible hospital admission in the study period (*n* = 2093, 3.6%), and lack of BPMH data (*n* = 3788, 6.5%). Based on eligibility for BPMH, 4113 admissions to one of the eligible AMUs, via ED, were further screened. Of these, 1825 patients were excluded using three additional predetermined criteria (no MedRec conducted, MedRec conducted beyond 48 h of ED transfer, or having incomplete data). MedRec was either not conducted in 40.2% (814 of 2027), 4.2% (27 of 645), and 4.7% (68 of 1441), or conducted beyond 48 h of ED transfer in 20.8% (421 of 2027), 23.1% (149 of 645), and 16.0% (231 of 1441) of patients in the PPMC, early BPMH, and usual care arms, respectively. The eligibility criteria for evaluation were met by 2288 patients, of whom 1048 randomly selected patients were included in the analysis: 230 in the PPMC group, 230 in the early BPMH group, and 588 in the usual care group.

Table 1 presents the demographic and clinical characteristics of the samples. Except for sex (*p* = 0.47), all other characteristics were significantly different between the groups. Patients in the PPMC group were comparatively older (median age, 78 years vs. 72 years; *p* < 0.001) and had more complex medical conditions (median CCI, 5 vs. 4; *p* < 0.001) than those in the comparison groups. A higher median number of medicines were also initially charted in the PPMC group than in the early BPMH group (*p* = 0.02) or the usual care group (*p* < 0.001). Patients admitted to the general medicine unit constituted the majority of the cohort: 203 (88%) in the PPMC group, 123 (53%) in the early BPMH group, and 220 (37%) in the usual care group. 

### 3.2. Prevalence and Types of Medication Discrepancies and Errors

A total of 22 discrepancies were identified in the PPMC group, compared to 360 in the early BPMH group and 1042 in the usual care group (Appendix A). The median number of discrepancies (0 vs. 1) and the median number of discrepancies per prescribed medication (0 vs. 0.1) were different between the PPMC group and the comparison groups (*p* < 0.001). “Omitted drug” was the most commonly identified discrepancy in all study groups, followed by “Different dose” and “Different frequency” (Table 2).

One hundred twenty-three cases containing 158 discrepancies were reviewed by the panel. Approximately 64%, 87%, and 88% of the reviewed discrepancies were deemed “unintentional medication errors” by the panel in the PPMC group, the early BPMH group, and the usual care group, respectively. Further panel assessment findings are provided in Appendix A. Fleiss’ kappa showed that there was a ‘fair’ agreement [35] between the panel members’ judgements for inferring the intentionality of discrepancies (κ = 0.25, *p* < 0.001) and rating the likely clinical severity of errors (κ = 0.22, *p* < 0.001). 

### 3.3. Patients Having at Least One Error

The PPMC group had a reduced prevalence of errors and high/extreme risk errors compared to the comparison groups (Figure 3). There were fewer patients who had at least one unintentional error in the PPMC group (3.5%, 95% CI: 1.1% to 5.8%) than in the early BPMH (49.4%, 95% CI: 42.5% to 56.3%) and usual care group (61.4%, 95% CI: 56.3% to 66.7%). Only 1.3% (95% CI: 0% to 2.8%) of the patients in the PPMC group had at least one high/extreme risk error, which was lower than the early BPMH group (23.2%, 95% CI: 14.4% to 32.1%) and the usual care group (26.6%, 95% CI: 18.6% to 34.6%). 

Compared to the early BPMH group, the NNTs with PPMC to prevent at least one additional error and one additional high/extreme risk error were 2.2 (95% CI: 2.0 to 2.4) and 4.6 (95% CI: 3.4 to 6.9); when compared to the usual care group, the corresponding NNTs were 1.7 (95% CI: 1.6 to 1.8) and 4.0 (95% CI: 3.1 to 5.3), respectively. Some case vignettes of errors bearing high or extreme risk are presented in Appendix A.

### 3.4. Number of Errors per Medication Order

There were 0.6 (95% CI: 0.3 to 0.8), 13.8 (95% CI: 12.3 to 15.2), and 17.2 (95% CI: 16.2 to 18.2) errors for every 100 medication orders for initially charted medicines in the PPMC group, the early BPMH group, and the usual care group, respectively (Appendix A). Comparatively, patients in the PPMC group had 0.1 high/extreme risk errors (95% CI: 0.0 to 0.3) for every 100 medication orders, whereas the corresponding figures in the early BPMH and usual care group were 5.9 (95% CI: 4.9 to 6.8) and 6.7 (95% CI: 6.0 to 7.3), respectively.

### 3.5. Errors Involving High-Risk and Time-Critical Medicines

No medication order for HRMs or TCMs had an error bearing high or extreme risk in the PPMC group (Appendix A). This contrasted with 7.4% (95% CI: 4.8% to 10%) and 11% (95% CI: 8.9% to 13.2%) of the medication orders for HRMs in the early BPMH group and the usual care group, respectively, that had errors bearing high/extreme risk. Similarly, 4.8% (95% CI: 2.7% to 6.9%) and 3.6% (95% CI: 2.2% to 4.9%) of the medication orders for TCMs were estimated to have errors bearing high/extreme risk in the early BPMH group and the usual care group, respectively.

## 4. Discussion

There were significant reductions in the prevalence of medication discrepancies and errors, and likely clinical severity of the errors with PPMC, when compared to either early BPMH alone or usual care. Moreover, the occurrences of errors per prescribed medication and errors involving HRMs or TCMs were significantly reduced when medications were charted using the PPMC model of care in the ED. Compared to usual care, documenting a BPMH, which was available to medical officers in the ED before charting, reduced the occurrences of discrepancies but did not affect the prevalence of errors or their clinical severity. 

Our study differs from prior PPMC studies [15,30] due to the inclusion of a third arm (early BPMH alone) for comparison with PPMC or usual care. This allowed us to compare the effect of early BPMH in isolation with the subsequent collaborative discussion that invariably occurred between a PPMC-credentialled pharmacist and a medical officer in the ED in the PPMC model. The distinction between PPMC and early BPMH alone, in terms of their relative effectiveness, likely stems from the PPMC’s collaborative clinical discussion. Evidence shows that the majority of medication error corrections may result from interprofessional discussions rather than through retrospective approaches, such as reviewing written medication orders [36,37,38]. PPMC was intended to utilise pharmacists’ expertise early in the ED by developing accurate medication histories and discussing treatment intentions with medical officers. By contrast, the early BPMH strategy alone did not improve key outcomes, such as occurrences of high/extreme errors, compared to usual care. A possible explanation for this includes the absence of the collaborative pharmacist-medical officer conversation and the subsequent development of an agreed medication treatment plan (i.e., the hallmarks of PPMC) both in the early BPMH and usual care approaches. For the best use of the skill mix within the ED patient care models, closer interprofessional collaboration is required.

Consistent with the findings of earlier collaborative charting studies [15,30,39,40], PPMC in this study was able to reduce medication errors prior to administration. Our findings were also comparable to a systematic review that reported that the largest reduction in the prevalence of medication errors tended to involve pharmacists in medication co-prescribing/co-charting [38]. This finding is important in the ED, where the medication use process is time-consuming and complex for several reasons. Most ED sources of medication information have several inconsistencies and cannot be relied on in isolation [41]. Overcrowding, increased workloads, inadequate infrastructure, urgency of care, heterogeneity of medications, and lack of familiarity with patients and their medications may play a role in increased rates of medication errors in ED [6,9,42,43,44]. Working as a team could reduce the workload of busy medical officers in the ED, who devote a relatively small percentage of their time to medication-related tasks, estimated at 18% [45]. 

### 4.1. Clinical Implications

The evidence from this study indicates that significantly more patients who received the PPMC had a medication chart that was free of errors. Obtaining and documenting accurate and complete patients’ medication histories early in the ED and optimising their initial medication regimen through a collaborative discussion between a pharmacist and medical officer are key in avoiding possible errors that might otherwise remain in the patient’s active medication list and perpetuate to inpatient wards. Reducing the prevalence of errors bearing high/extreme risk, which could otherwise lead to hospital-acquired complications or lengthened hospital stays, was the important clinical implication of PPMC. 

Following the conclusion of the pilot project on 17 May 2021, the PPMC service was integrated into the hospital’s regular pharmacy operations and has been adopted more widely locally. Future PPMC initiatives may consider expanding PPMC to cover additional clinical areas and working hours. However, when scaling up PPMC, efforts should be made to tailor the PPMC credentialling to the needs, procedures, and policies of different clinical settings [46].

### 4.2. Strengths and Limitations

This study’s strength was a simultaneous comparison of the impact of three ED-based care models on medication discrepancies and errors using a relatively large sample size and long study period. In addition, a robust panel assessment approach was followed, which included thorough independent case preparation and review, and several rounds of blinded, independent, interdisciplinary expert panel assessments. External, independent researchers conducted this evaluation study to avoid organisational bias.

Being a pragmatic evaluation trial conducted in a real practice setting, there are some limitations to consider. Possible biases may exist because the study was not designed as a prospective randomised controlled trial. ED staff pragmatically assigned patients to the study groups in a non-random fashion, which may have led to a selection bias. As shown in Table 1, the baseline characteristics were different between the groups, with patients in the PPMC group being comparatively older, with more medically complex conditions and a higher number of medications than those in the comparison groups. The effect of PPMC may have been even more pronounced, however, if the study participants’ characteristics in the three arms had been more similar.

Patients who were excluded from the study were predominantly children (<18 years), those not subsequently admitted, those admitted to units other than an eligible AMU, overnight ED presentations, not being the first eligible admission, and lack of BPMH data (Figure 2). PPMC service was also limited to ED patients with intended admissions to either a general medicine, emergency medicine, or mental health unit during pharmacists’ working hours, accounting for the relatively low participation. Due to these reasons, only a limited number of patients were potentially eligible for evaluation. Patients who did not receive MedRec within 48 h of ED transfer were also excluded, allowing us to compare the medication charting issues between the study groups. Presumably, because they perceived that charts written using the PPMC model had already undergone a safety check in ED, ward pharmacists were less likely to conduct an inpatient MedRec for every PPMC patient, resulting in more study exclusions for that group. This implies poor MedRec practice on the wards and needs to be improved to ensure the continuity of medication management and safety at all stages of hospitalisation. 

Reviewing medication discrepancies and errors from routinely collected data is subject to limitations. When collecting data in retrospect, there is a possibility that not all the information is recorded completely and accurately. Neither the ED’s nor the medical officers’ busyness was measured at the time of charting. Patients were easily identifiable during the data collection, making blinding the data collector impossible. 

When evaluating the clinical significance of the cases, there was a ‘fair’ degree of agreement across the assessors. Although having a wide range of experts with multiple perspectives is important for the method’s robustness, the different clinical backgrounds and experiences of the assessors may explain the observed degree of agreement. Additionally, the nature of the rating scale can offer varied perspectives, hence influencing inter-assessor agreement [47]. The risk rating was also based on a possible event in the future, which was irrespective of the time. Some of the events could potentially occur in the distant future; however, in the meantime, the error could be corrected by other clinicians, such as general practitioners and community pharmacists. 

## 5. Conclusions

There was strong evidence that PPMC was beneficial in reducing the occurrence of clinically significant medication errors when compared to early BPMH alone or usual care in the ED. Obtaining an early BPMH alone and providing this to the medical officer was not associated with improved key outcomes, highlighting the importance of the co-charting process. Our findings highlight the importance of closer interprofessional collaborative medication charting practices in the ED.

## Figures and Tables

**Figure 1 ijerph-20-01452-f001:**
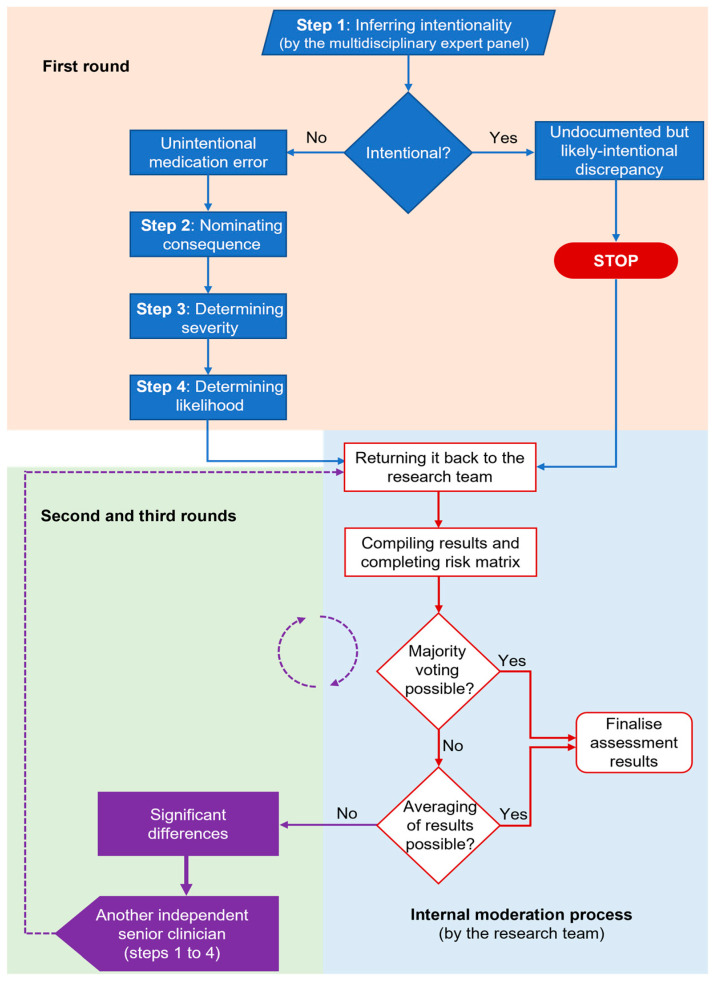
The panel assessment rounds and internal moderation process.

**Figure 2 ijerph-20-01452-f002:**
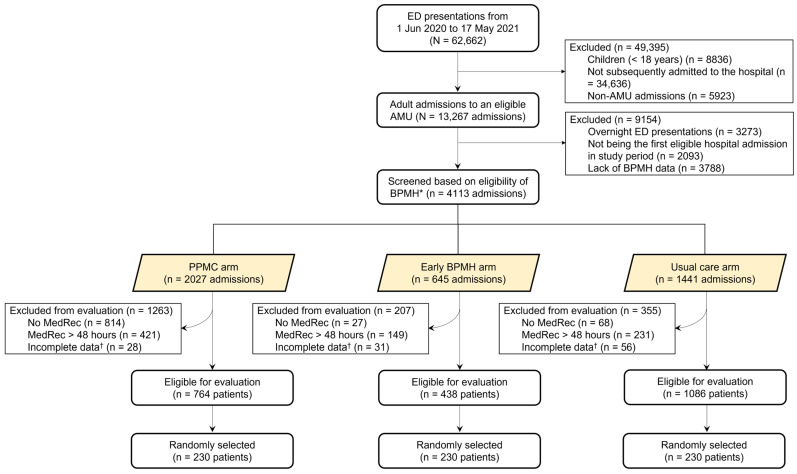
Diagram illustrating the selection of study participants. Note: Abbreviations: AMU, acute medical unit; BPMH, best-possible medication history; ED, emergency department; MedRec, medication reconciliation; PPMC, partnered pharmacist medication charting. * BPMH was limited to within 48 h post-admission (i.e., until the MedRec time) in the usual care arm. † Examples include incomplete/unavailable discharge summary or medication chart information.

**Figure 3 ijerph-20-01452-f003:**
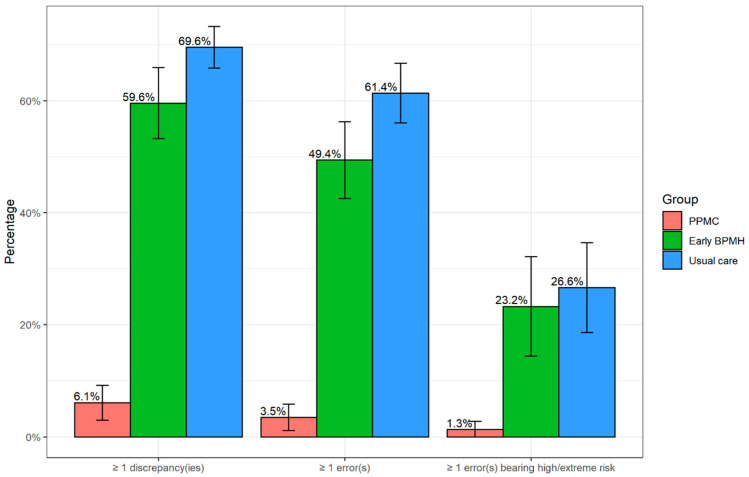
Patients who had at least one discrepancy/error and error bearing high/extreme risk. Abbreviations: BPMH, best-possible medication history; PPMC, partnered pharmacist medication charting. Error bars denote 95% confidence intervals.

**Table 1 ijerph-20-01452-t001:** Demographic and clinical characteristics of study participants.

Variables	Study Group			*p*-Value	
	PPMC (N = 230 Patients)	Early BPMH (N = 230 Patients)	Usual Care (N = 588 Patients)	Overall	Pairwise
Sex female, n (%)	122 (53%)	111 (48%)	310 (53%)	0.47 *	
Age in years, median (IQR)	79 (68, 87)	72 (59, 82)	72 (62, 81)	**<0.001** ^†^	**<0.001** ^‡,§^, 0.93 ^¶^
<65 years, n (%)	44 (19%)	80 (35%)	189 (32%)	**<0.001** *	
≥65 years, n (%)	186 (81%)	150 (65%)	399 (68%)		
CCI, median (IQR)	5 (4, 6)	4 (3, 6)	4 (3, 5.3)	**<0.001** ^†^	**<0.001** ^‡,§^, 0.93 ^¶^
ATS, median (IQR)	3 (3, 4)	3 (2, 4)	3 (2, 3)	**0.002** ^†^	**0.003** ^‡^, 0.45 ^§^, **0.03** ^¶^
Medicines, median (IQR)					
Preadmission medicines	10 (7, 14)	9 (5, 13)	9 (6, 13)	**0.02** ^†^	**0.02** ^‡^, **0.04** ^§^, 0.95 ^¶^
Initially charted medicines	10 (8, 14)	10 (7, 12)	9 (6, 12)	**<0.001** ^†^	**<0.001** ^‡^, **0.02** ^§^, **0.03** ^¶^
Acute admission units, n (%)				**<0.001** ‖	
Cardiology	0 (0%)	16 (7.0%)	67 (11%)		
Emergency Medicine	27 (12%)	40 (17%)	112 (19%)		
General Medicine	203 (88%)	123 (53%)	220 (37%)		
Psychiatry	0 (0%)	2 (0.9%)	15 (2.6%)		
Respiratory Medicine	0 (0%)	8 (3.5%)	56 (9.5%)		
Stroke	0 (0%)	13 (5.7%)	67 (11%)		
Others #	0 (0%)	28 (12%)	51 (8.7%)		
Time from ED arrival (hours)					
BPMH, median (IQR)	4 (3, 6)	4 (3, 6)	25 (20, 42)	**<0.001** ^†^	**<0.001**^‡,¶^, 0.79 ^§^
MedRec, median (IQR)	24 (21, 29)	21 (11, 30)	27 (21, 44)	**<0.001** ^†^	**<0.001** ^‡,§,¶^

Abbreviations: ATS, Australasian Triage Scale; BPMH, best-possible medication history; CCI, Charlson comorbidity index; IQR, interquartile range; n, number; PPMC, partnered pharmacist medication charting. * Pearson’s Chi-square test. ^†^ Kruskal-Wallis with Dunn’s post-hoc test: ^‡^ PPMC vs. usual care; ^§^ PPMC vs. early BPMH; ^¶^ early BPMH vs. usual care. ‖ Fisher’s exact test. # Others: Endocrinology, Renal Medicine, Rheumatology. Bold numbers indicate statistically significant findings.

**Table 2 ijerph-20-01452-t002:** Types of undocumented medication discrepancies.

Outcome	Stud Group		
	PPMC (N = 230 Patients)	Early BPMH (N = 230 Patients)	Usual Care (N = 588 Patients)
Total discrepancies	22	360	1042
Discrepancy types, n (% of discrepancies)			
Omitted drug *	18 (81.8%)	247 (68.6%)	673 (64.6%)
Different drug	0 (0%)	11 (3.1%)	20 (1.9%)
Added drug	0 (0%)	11 (3.1%)	38 (3.6%)
Different dose ^†^	2 (9.1%)	44 (12.2%)	160 (15.4%)
Different frequency ^†^	2 (9.1%)	32 (8.9%)	105 (10.1%)
Different route	0 (0%)	0 (0%)	3 (0.3%)
Different dosage form	0 (0%)	0 (0%)	10 (1.0%)
Incomplete order	0 (0%)	15 (4.2%)	33 (3.2%)

Abbreviations: BPMH, best-possible medication history; n, number; PPMC, partnered pharmacist medication charting. * Included medications charted in the wrong chart type or section, e.g., insulin not charted in the insulin chart (*n* = 1 in the early BPMH group and *n* = 4 in the usual care group). ^†^ Each of the 12 medications in the early BPMH group and 41 medications in the usual care group had two discrepancies: a different dose and a different frequency.

## Data Availability

All data generated or analysed during this study are included in this published article [and its Appendix A].

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
