# Peer review of "Impact of Partnered Pharmacist Medication Charting (PPMC) on Medication Discrepancies and Errors: A Pragmatic Evaluation of an Emergency Department-Based Process Redesign"

_ijerph, 2023, doi:10.3390/ijerph20021452_

Round 1

Reviewer 1 Report

Pragmatic study design could potentially have led to bias - should be noted as a limitation of the study. Did the ED staff allocate those with more complex medication regimens to the PPMC arm? 

Study participant numbers are quite small compared to those eligible. The reasons for this should be explained in the discussion.

The difference between this study and previous PPMC studies is the comparison against early BPMH. This could be made clearer in the paper.

Author Response

Version 1 – Authors’ response to reviewers’ comments

Title

Impact of partnered pharmacist medication charting (PPMC) on

medication discrepancies and errors: A pragmatic evaluation of an emergency department-based process redesign

Manuscript ID

ijerph-2131884

Response to Reviewer 1 Comments

Point 1: Pragmatic study design could potentially have led to bias - should be noted as a limitation of the study. Did the ED staff allocate those with more complex medication regimens to the PPMC arm? 

Response 1: As suggested, we have further acknowledged the possibility of bias with pragmatic trials in our revised manuscript.

Being a pragmatic evaluation trial conducted in a real practice setting, there are some limitations to consider. Possible biases may exist because the study was not designed as a prospective randomised controlled trial. ED staff pragmatically assigned patients to the study groups in a non-random fashion, which may have led to a selection bias. As shown in Table 1, the baseline characteristics were different between the groups, with patients in the PPMC group being comparatively older, with more medically complex conditions and a higher number of medications than those in the comparison groups. The effect of PPMC may have been even more pronounced, however, if the study participants’ characteristics in the three arms had been more similar.” [Lines 430–438]

Point 2: Study participant numbers are quite small compared to those eligible. The reasons for this should be explained in the discussion.

Response 2: Thanks for asking us to clarify this point. In our revised manuscript, we have now included the following exclusions (underlined ones) in 3.1. Patient characteristics. Figure 2 has also been amended to reflect this.

“Between June 1, 2020, and May 17, 2021, a total of 62,662 patients presented to the RHH ED (Figure 2). Of these, 58,549 presentations were excluded using predetermined criteria: children under the age of 18 (n = 8,836, 15.1%), those not subsequently admitted to the hospital (n = 34,636, 59.2%), those admitted to units other than an eligible AMU (n = 5,923, 10.1%), overnight ED presentations (n = 3,273, 5.6%), not being the first eligible hospital admission in the study period (n = 2,093, 3.6%) and lack of BPMH data (n = 3,788, 6.5%). On the basis of eligibility for BPMH, 4,113 admissions to one of the eligible AMUs, via ED, were further screened. Of these, 1,825 patients were excluded using three additional predetermined criteria (no MedRec conducted, MedRec conducted beyond 48 hours of ED transfer or having incomplete data). MedRec was either not conducted in 40.2% (814 of 2027), 4.2% (27 of 645) and 4.7% (68 of 1,441), or conducted beyond 48 hours of ED transfer in 20.8% (421 of 2027), 23.1% (149 of 645) and 16.0% (231 of 1,441) of patients in the PPMC, early BPMH and usual care arms, respectively. The eligibility criteria for evaluation were met by 2,288 patients, of whom 1,048 randomly selected patients were included in the analysis: 230 in the PPMC group, 230 in the early BPMH group and 588 in the usual care group.” [Lines 275–290]

Around half of the eligible ones were included in the analysis. The number of study participants was estimated based on a power calculation (229 per study group), as shown in the manuscript’s section 2.9 (lines 265–272).

In addition, we have now included the following paragraph in our revised manuscript’s discussion section to provide commentary on the exclusion criteria.

“Patients who were excluded from the study were predominantly children (< 18 years), those not subsequently admitted, those admitted to units other than an eligible AMU, overnight ED presentations, not being the first eligible admission and lack of BPMH data (Figure 2). PPMC service was also limited to ED patients with intended admissions to either a general medicine, emergency medicine or mental health unit during pharmacists’ working hours, accounting for the relatively low participation. Due to these reasons, only a limited number of patients were potentially eligible for evaluation.” [Lines 439–445]

Point 3: The difference between this study and previous PPMC studies is the comparison against early BPMH. This could be made clearer in the paper.

Response 3: As suggested, we have now added the following sentences (underlined ones) in our revised manuscript.

Our study differs from prior PPMC studies [1, 2] due to the inclusion of a third arm (early BPMH alone) for comparison with PPMC or usual care. This allowed us to compare the effect of early BPMH in isolation with the subsequent collaborative discussion that invariably occurred between a PPMC-credentialled pharmacist and a medical officer in ED in the PPMC model. The distinction between PPMC and early BPMH alone, in terms of their relative effectiveness, likely stems from the PPMC’s collaborative clinical discussion. Evidence shows that the majority of medication error corrections may result from interprofessional discussions rather than through retrospective approaches, such as reviewing written medication orders [3-5]. PPMC was intended to utilise pharmacists’ expertise early in the ED by developing accurate medication histories and discussing treatment intentions with medical officers.[Lines 378–388]

The following was already included in our original submission.

This study aimed to evaluate the impact of PPMC on the prevalence of medication discrepancies and errors, and their clinical significance. In addition, this study investigated whether obtaining and documenting an early best-possible medication history (BPMH) in ED, without PPMC, could reduce medication discrepancies and errors compared to usual care.” [Lines 55–59]

References used in the authors’ response

  1. Tong, E. Y.; Roman, C.; Mitra, B.; Yip, G.; Gibbs, H.; Newnham, H.; Smit, D. P.; Galbraith, K.; Dooley, M. J., Partnered pharmacist charting on admission in the General Medical and Emergency Short-stay Unit - a cluster-randomised controlled trial in patients with complex medication regimens. J. Clin. Pharm. Ther. 2016, 41, (4), 414–418.
  2. Tong, E. Y.; Mitra, B.; Yip, G.; Galbraith, K.; Dooley, M. J.; Group, P. R., Multi-site evaluation of partnered pharmacist medication charting and in-hospital length of stay. Br. J. Clin. Pharmacol. 2020, 86, (2), 285–290.
  3. Patanwala, A. E.; Sanders, A. B.; Thomas, M. C.; Acquisto, N. M.; Weant, K. A.; Baker, S. N.; Merritt, E. M.; Erstad, B. L., A prospective, multicenter study of pharmacist activities resulting in medication error interception in the emergency department. Ann. Emerg. Med. 2012, 59, (5), 369–373.
  4. Weeks, G. R.; Ciabotti, L.; Gorman, E.; Abbott, L.; Marriott, J. L.; George, J., Can a redesign of emergency pharmacist roles improve medication management? A prospective study in three Australian hospitals. Res. Social Adm. Pharm. 2014, 10, (4), 679–92.
  5. Atey, T. M.; Peterson, G. M.; Salahudeen, M. S.; Bereznicki, L. R.; Wimmer, B. C., Impact of pharmacist interventions provided in the emergency department on quality use of medicines: a systematic review and meta-analysis. Emerg. Med. J. 2022, emermed–2021–211660.

Reviewer 2 Report

This is a well designed pragmatic study that presents a single institutional experience with a evolving model of care

Areas that should be considered and expanded include:

1)  commentary of exclusion of patients that did not receive a MedRec within 48hours ...ie medically eligible patients but only excluded for this reason... what was the magnitude of this group and as a result ..any implications

2) has this model been adopted more widely locally and commentary as to whether should be expanded to other medical units 

3) additional commentary in the discussion re Beks, H., Namara, K.M., Manias, E. et al. Hospital pharmacists’ experiences of participating in a partnered pharmacist medication charting credentialing program: a qualitative study. BMC Health Serv Res 21, 251 

4) Line 270: " reasons given in Figure 2 " with no detail provided in figure 2

5) Additional commentary required re lack of difference between early BPMH and usual care and implications in informing future practice

6) Further commentary re 4,113 of 62662 ED presentation eligible and implication of expansion of PPMC model

Author Response

Version 1 – Authors’ response to reviewers’ comments

Title

Impact of partnered pharmacist medication charting (PPMC) on

medication discrepancies and errors: A pragmatic evaluation of an emergency department-based process redesign

Manuscript ID

ijerph-2131884

Response to Reviewer 2 Comments

Point 1: This is a well designed pragmatic study that presents a single institutional experience with a evolving model of care

Response 1: Thank you for the positive evaluation.

Point 2: commentary of exclusion of patients that did not receive a MedRec within 48hours ...ie medically eligible patients but only excluded for this reason... what was the magnitude of this group and as a result ..any implications

Response 2: As suggested, we have now discussed it further, as detailed below. The underlined sentences below are new additions to our revised manuscript.

“.. Patients who did not receive MedRec within 48 hours of ED transfer were also excluded, allowing us to compare the medication charting issues between the study groups. Presumably, because they perceived that charts written using the PPMC model had already undergone a safety check in ED, ward pharmacists were less likely to conduct an inpatient MedRec for every PPMC patient, resulting in more study exclusions for that group. This implies poor MedRec practice on the wards and needs to be improved to ensure the continuity of medication management and safety at all stages of hospitalisation.” [Lines 445–452]

Point 3: has this model been adopted more widely locally and commentary as to whether should be expanded to other medical units 

Response 3: As suggested, our revised manuscript now includes the following lines.

“Following the conclusion of the pilot project on 17 May 2021, the PPMC service was integrated into the hospital’s regular pharmacy operations and has been adopted more widely locally. Future PPMC initiatives may consider expanding PPMC to cover additional clinical areas and working hours. However, when scaling up PPMC, efforts should be made to tailor the PPMC credentialling to the needs, procedures and policies of different clinical settings [1].” [Lines 417–422]

Point 4: additional commentary in the discussion re Beks, H., Namara, K.M., Manias, E. et al. Hospital pharmacists’ experiences of participating in a partnered pharmacist medication charting credentialing program: a qualitative study. BMC Health Serv Res 21, 251 

Response 5: Based on the comment, we have now added the following line in our revised manuscript.

“… However, when scaling up PPMC, efforts should be made to tailor the PPMC credentialling to the needs, procedures and policies of different clinical settings [1].” [Lines 420–422]

Point 5: Line 270: " reasons given in Figure 2 " with no detail provided in figure 2

Response 5: As suggested, we have now updated Figure 2 to provide the reasons for exclusion. [Lines 282–287]

We have also included the following in the revised manuscript’s Results section: “MedRec was either not conducted in 40.2% (814 of 2027), 4.2% (27 of 645) and 4.7% (68 of 1,441), or conducted beyond 48 hours of ED transfer in 20.8% (421 of 2027), 23.1% (149 of 645) and 16.0% (231 of 1,441) of patients in the PPMC, early BPMH and usual care arms, respectively.” (Lines 284–287)

Figure 2. Diagram illustrating the screening and selection of study participants.

Abbreviations: AMU, acute medical unit; BPMH, best-possible medication history; ED, emergency department; MedRec, medication reconciliation; PPMC, partnered pharmacist medication charting.

*BPMH was limited to within 48 hours post-admission (i.e. until the MedRec time) in the usual care arm.

†Examples include incomplete/unavailable discharge summary or medication chart information.

Point 6: Additional commentary required re lack of difference between early BPMH and usual care and implications in informing future practice

Response 6: As suggested, we have now added the following line in our revised manuscript.

“… By contrast, the early BPMH alone strategy did not improve key outcomes, such as occurrences of high/extreme errors, compared to usual care. A possible explanation for this includes the absence of the collaborative pharmacist-medical officer conversation and the subsequent development of an agreed medication treatment plan (i.e. the hallmarks of PPMC) both in the early BPMH and usual care approaches. For the best use of the skill mix within the ED patient care models, closer interprofessional collaboration is required.[Lines 388–394]

Point 7: Further commentary re 4,113 of 62662 ED presentation eligible and implication of expansion of PPMC model

Response 7: In response to this suggestion, we have now included the following sentences in the revised manuscript. Figure 2 provides further exclusion details.

“Patients who were excluded from the study were predominantly children (< 18 years), those not subsequently admitted, those admitted to units other than an eligible AMU, overnight ED presentations, not being the first eligible admission and lack of BPMH data (Figure 2). PPMC service was also limited to ED patients with intended admissions to either a general medicine, emergency medicine or mental health unit during pharmacists’ working hours, accounting for the relatively low participation. Due to these reasons, only a limited number of patients were potentially eligible for evaluation.” [Lines 439–444]

“Following the conclusion of the pilot project on 17 May 2021, the PPMC service was integrated into the hospital’s regular pharmacy operations and has been adopted more widely locally. Future PPMC initiatives may consider expanding PPMC to cover additional clinical areas and working hours. However, when scaling up PPMC, efforts should be made to tailor the PPMC credentialling to the needs, procedures and policies of different clinical settings [1].” [Lines 417–422]

References used in the authors’ response

  1. Beks, H.; Namara, K. M.; Manias, E.; Dalton, A.; Tong, E.; Dooley, M., Hospital pharmacists' experiences of participating in a partnered pharmacist medication charting credentialing program: a qualitative study. BMC health services research 2021, 21, (1), 251.

Reviewer 3 Report

46-47: Kindly avoid ambiguity as it’s not a top health priority in Australia. It’s a 10th National Health Priority.

65: Provide a direct web link to verify numbers – the current link will take the reader to a generic hospital page only.

74: How were they trained for the provision of PPMC?

161: Include definition for ‘time-critical medicine’.

172: UTAS was not earlier defined.

149: PRN was also defined later. It’s a Latin phrase and not the right abbreviation for ‘as needed’.

197: Define SHPA first.

229: The shapiro-Wilk test is more appropriate for smaller samples (<50).

368: Reference 37 should not have been cited as it was a review article - Apparently an inappropriate self-citation. All other references were original research articles.

379: Not clear enough. How did it improve the medication chart accuracy? was it included as one of the study outcomes? Not sure how you reached this outcome because there were also significant differences in the baseline characteristics of all included groups.

382: collaborative discussion between? - Seems vague.

Author Response

Version 1 – Authors’ response to reviewers’ comments

Title

Impact of partnered pharmacist medication charting (PPMC) on

medication discrepancies and errors: A pragmatic evaluation of an emergency department-based process redesign

Manuscript ID

ijerph-2131884

Response to Reviewer 3 Comments

Point 1: 46-47: Kindly avoid ambiguity as it’s not a top health priority in Australia. It’s a 10th National Health Priority.

Response 1: Thank you for the suggestion, but there are only ten health priority areas and medicine safety and quality use of medicines is one of them.

We have amended this in the revised manuscript.

It is worth noting that the Australian Health Ministers recently agreed to recognise medicine safety and quality use of medicines as one of the national health priority areas.” [Lines 45–47]

Point 2: 65: Provide a direct web link to verify numbers – the current link will take the reader to a generic hospital page only.

Response 2: We have now updated the citation link (given below).

https://www.health.tas.gov.au/hospitals/royal-hobart-hospital/about-royal-hobart-hospital [Lines 551–552]

Point 3: 74: How were they trained for the provision of PPMC?

Response 3: In our original submission, we already included the following sentences to shed light on the credentialling process.

“The following were PPMC-credentialling pathway requirements: (a) completion of all prior reading and education modules, (b) clinical assessment via the Society of Hospital Pharmacists of Australia (SHPA)’s Clinical Competency Achievement Tool (ClinCAT®) [1], (c) observation of one PPMC clinical episode, (d) completion of five PPMC clinical episodes under the supervision of a PPMC-credentialled pharmacist or a medical officer, (e) completion of a PPMC-Training Objective Structured Clinical Examination with a supervising pharmacist and a medical consultant, and (f) formal recognition by the THS Statewide Credentialling Committee.” [Lines 74–83]

We have also included the following new information in our revision. “ClinCAT® is a competency achievement tool that supports Australian pharmacists’ training and development. Accredited by the Australian Pharmacy Council, ClinCAT® includes activities defined by SHPA’s standards of practice for clinical pharmacy services [2].” [Lines 83–86]

Point 4: 161: Include definition for ‘time-critical medicine’.

Response 4: The following definition of time-critical medicines was already given in our original manuscript.

TCMs are defined as “medicines where early or delayed administration by more than 30 minutes from the prescribed time for administration may cause harm to the patient or compromise the therapeutic effect” [3].” [Lines 172–175]

Point 5: 172: UTAS was not earlier defined.

Response 5:: Thanks for noting this. We have now fixed it in our revision. [Line 67]

Point 6: 149: PRN was also defined later. It’s a Latin phrase and not the right abbreviation for ‘as needed’.

Response 6: We have now defined it in our revision. [Line 156]

Point 7: 197: Define SHPA first.

Response 7: We have now defined it in our revision. [Line 78]

Point 8: 229: The Shapiro-Wilk test is more appropriate for smaller samples (<50).

Response 8: Yes, we agree with the reviewer that the Shapiro-Wilk test is more appropriate when there are smaller sample sizes. However, the test can handle larger sample sizes as well and even some statisticians claim the Kolmogorov-Smirnov test has lower statistical power than the Shapiro-Wilk test [4, 5]. Additionally, we employed graphical methods to check for the normality of our data. To our understanding, having a larger sample size does not preclude us from using the Shapiro-Wilk test for normality checking. Even with the Kolmogorov-Smirnov test, we found that the interpretation of the Shapiro-Wilk test’s normality analysis remained unchanged.

Point 9: 368: Reference 37 should not have been cited as it was a review article - Apparently an inappropriate self-citation. All other references were original research articles.

Response 9: As suggested, we have removed the citation from that line and now appropriately cited the review article in our revised manuscript. “Our findings were also comparable to a systematic review that reported that the largest reduction in the prevalence of medication errors tended to involve pharmacists in medication co-prescribing/co-charting [6].” [Lines 396–398]

Point 10: 379: Not clear enough. How did it improve the medication chart accuracy? was it included as one of the study outcomes? Not sure how you reached this outcome because there were also significant differences in the baseline characteristics of all included groups.

Response 10: We meant “medication chart accuracy” to refer to the proportion of error-free medication charts. Based on the comment, we have now reworked the sentence in our revised manuscript as below.

The evidence from this study indicated that significantly more patients who received the PPMC had a medication chart that was free of errors.” [Lines 409–410]

Point 11: 382: collaborative discussion between? - Seems vague.

Response 11: We have now clarified this by adding the following one in our revision:, between a pharmacist and medical officer,”. [Lines 412–413]

References used in the authors’ response

  1. Society of Hospital Pharmacists of Australia (SHPA) SHPA ClinCAT. Available online: https://www.shpa.org.au/cpd/shpa-clincat#:~:text=The%20ClinCAT%C2%AE%20achievement%20tool,equip%20them%20to%20perform%20evaluations. [cited 2022 August 13].
  2. Society of Hospital Pharmacists of Australia (SHPA) Standards of Practice for Clinical Pharmacy Services. Available online: https://www.shpa.org.au/publications-resources/standards-of-practice/standards-of-practice-for-clinical-pharmacy-services [cited 2023 January 7].
  3. Government of Western Australia Department of Health, Guiding Principles for Timely Administration of Medications. . In 2020.
  4. Mishra, P.; Pandey, C. M.; Singh, U.; Gupta, A.; Sahu, C.; Keshri, A., Descriptive statistics and normality tests for statistical data. Ann. Card. Anaesth. 2019, 22, (1), 67.
  5. Yap, B. W.; Sim, C. H., Comparisons of various types of normality tests. Journal of Statistical Computation and Simulation 2011, 81, (12), 2141-2155.
  6. Atey, T. M.; Peterson, G. M.; Salahudeen, M. S.; Bereznicki, L. R.; Wimmer, B. C., Impact of pharmacist interventions provided in the emergency department on quality use of medicines: a systematic review and meta-analysis. Emerg. Med. J. 2022, emermed–2021–211660.
